# Immune Checkpoint Inhibitor, Nivolumab, Combined with Chemotherapy Improved the Survival of Unresectable Advanced and Metastatic Esophageal Squamous Cell Carcinoma: A Real-World Experience

**DOI:** 10.3390/ijms24087312

**Published:** 2023-04-15

**Authors:** Ming-Wei Kao, Yao-Hung Kuo, Kun-Chou Hsieh, Ching-Tai Lee, Shih-Chi Wu, Wen-Chi Yang

**Affiliations:** 1Division of Thoracic Surgery, Department of Surgery, E-DA Hospital, Kaohsiung 824, Taiwan; 2Faculty of School of Medicine, College of Medicine, I-Shou University, Kaohsiung 824, Taiwan; 3Department of Radiation Oncology, E-DA Hospital, Kaohsiung 824, Taiwan; 4Department of Gastroenterology, E-DA Hospital, Kaohsiung 824, Taiwan; 5Trauma and Emergency Center, China Medical University Hospital, Taichung 404, Taiwan; 6Graduate Institute of Clinical Medical Science, China Medical University College of Medicine, Taichung 404, Taiwan; 7Division of Hematology and Medical Oncology, Department of Internal Medicine, Pingtung Christian Hospital, Pingtung 900, Taiwan

**Keywords:** nivolumab, ipilimumab, immune checkpoint inhibitor, esophageal SCC

## Abstract

Patients with advanced esophageal squamous cell carcinoma (SCC) have a poor prognosis when treated with standard chemotherapy. Programmed death ligand 1 (PD-L1) expression in esophageal cancer has been associated with poor survival and more advanced stage. Immune checkpoint inhibitors, such as PD-1 inhibitors, showed benefits in advanced esophageal cancer in clinical trials. We analyzed the prognosis of patients with unresectable esophageal SCC who received nivolumab with chemotherapy, dual immunotherapy (nivolumab and ipilimumab), or chemotherapy with or without radiotherapy. Patients who received nivolumab with chemotherapy had a better overall response rate (ORR) (72% vs. 66.67%, *p* = 0.038) and longer overall survival (OS) (median OS: 609 days vs. 392 days, *p* = 0.04) than those who received chemotherapy with or without radiotherapy. In patients receiving nivolumab with chemotherapy, the duration of the treatment response was similar regardless of the treatment line they received. According to clinical parameters, liver and distant lymph nodes metastasis showed a trend of negative and positive impacts, respectively, on treatment response in the whole cohort and in the immunotherapy-containing regimen cohort. Nivolumab add-on treatment showed less gastrointestinal and hematological adverse effects, compare with chemotherapy. Here, we showed that nivolumab combined with chemotherapy is a better choice for patients with unresectable esophageal SCC.

## 1. Introduction

Esophageal cancer (EC) is the sixth most common cancer and the sixth most common cause of mortality among all cancers worldwide, with an estimated 604,100 new cases (3.1% of all cancers) and 544,076 cancer deaths (5.5% of all cancer deaths) in 2020 [1]. The incidence of esophageal cancer is highest in eastern Asia, followed by southern and eastern Africa [1]. There are two major types of esophageal cancer: esophageal squamous cell carcinoma (ESCC) and esophageal adenocarcinoma (EAC). Asian and African people have a higher incidence of ESCC; China, Japan, and southeast Africa accounting for 90% of all cases [2,3,4], while Western people have a higher incidence of EAC [5]. According to the Taiwan Ministry of Health and Welfare National Health Cancer Registration Report, the incidence of esophageal cancer is 22.02/100,000 in males and 1.68/100,000 in females, with 92.75% and 84.42% of ESCC occurring in males and females, respectively, in 2018.

Chemotherapy with or without radiotherapy is the standard treatment for unresectable advanced/recurrent ESCC. Combination therapy with platinum and 5FU is the major treatment in the first-line setting. Taxanes and other drugs are used as second-line therapies in patients who are relapse or refractory to first-line therapies. However, the response rate remained poor [6].

Tumor escape from antitumor immunity is critical for tumor survival and progression. Programmed cell death 1 (PD-1) is an immunosuppressive co-stimulatory signal receptor that belongs to the CD28 family. PD-1 was first identified by Ishida et al. in 1992 as a programmed cell death-induced gene encoding type I membrane proteins in T cells [7]. In esophageal cancer, elevations in PD-L1 and PD-L2 have been reported [8,9]. Increased PD-L1 expression is associated with poor survival and greater depth of tumor invasion [8,9]. In phase III clinical trials, ESCC patients who progressed after one prior therapy with the PD-1 inhibitor, pembrolizumab, showed improved overall survival (OS) to a median OS of 8.2 months, compared with 7.1 months in patients who received chemotherapy (KEYNOTE-181) [10]. ESCC patients who were refractory or intolerant to prior fluoropyrimidine-based and platinum-based chemotherapy received nivolumab and showed increased OS (nivolumab vs. paclitaxel/docetaxel: 10.9 months vs. 8.4 months) [11]. Based on the results of the KEYNOTE-590 and CheckMate-648 trials, the US Food and Drug Administration (FDA) approved that immune checkpoint inhibitors, pembrolizumab and nivolumab combined with chemotherapy and dual immunotherapy, ipilimumab-nivolumab, treat ESCC patients as first-line therapy, regardless of PD-L1 expression [12,13,14].

Here, we retrospectively analyzed the real-world data of treatment response and survival of patients with metastatic unresectable ESCC treated with nivolumab combined with chemotherapy, compared with dual immune checkpoint inhibitors, and standard chemotherapy.

## 2. Results

### 2.1. Patients’ Characteristics

Patient characteristics are shown in Table 1. There were no differences in age, sex, location of esophageal cancer, and comorbidities (diabetes, hypertension, liver cirrhosis, hepatitis B or C, and renal function) between nivolumab with chemotherapy (group 1), double checkpoint inhibitors (group 2), and chemotherapy groups (group 3). Patients in groups 1 and 2 showed a higher percentage of recurrent diseases when enrolled in the study (*p* = 0.045, Table 1). They also showed a more advanced stage (*p* = 0.020, Table 1), and more distant metastasis (*p* = 0.004, Table 1). In contrast, patients in the chemotherapy group had a related advanced T stage (*p* = 0.075, Table 1) and poor performance status (ECOG).

Patients using the immunotherapy-containing regimen could receive more lines of therapy (*p* = 0.008, Table 1). In ten patients who received first-line nivolumab and PF, one showed very long-term disease-free survival after six cycles of treatment. Three patients received nivolumab, PF, and radiotherapy as first-line treatment. One patient received cetuximab and PF as second-line therapy. Three patients who did not receive radiotherapy chose concurrent chemoradiotherapy with PF (CCRT) as second-line therapy. One of them had docetaxel and PF as third-line therapies. In twelve patients who received second-line nivolumab and chemotherapy, only one received cetuximab and PF as first-line therapy. All other patients received CCRT as first-line therapy. Three of them received nivolumab with paclitaxel and platinum as second-line therapy. The other nine patients received nivolumab and PF as second-line therapy. The other three patients who received nivolumab and PF as third-line therapy, received CCRT and docetaxel and PF as first- and second-line treatments, respectively.

In group 2, five of seven patients received CCRT as second-line therapy. Two patients had docetaxel and PF as third-line therapy. One of these two patients received six lines of therapy.

In group 3, two patients received docetaxel and PF as second-line therapy, one received gemcitabine, and one received platinum as second-line therapy due to poor performance status. The detailed treatments are shown in Appendix A.

### 2.2. Patients Who Received Nivolumab Combined with Chemotherapy Had a Trend of Longer Overall Survival and a Better Response than Those Who Received Standard Chemotherapy with or without Radiotherapy

In our cohort, we noted that patients receiving nivolumab and chemotherapy had a better overall response rate (ORR) than those who received chemotherapy, with or without radiotherapy (72% vs. 66.67%, *p* = 0.038, Table 1) and one fifth of patients receiving nivolumab and chemotherapy reached complete remission (CR). On the other hand, the majority of the patients in group 3 only received one line treatment (66.67%, Table 1), and most of the patients in group 1 received more than two lines treatment (*p* = 0.008, Table 1)

Patients who received nivolumab and chemotherapy showed the longest OS, followed by patients receiving nivolumab and ipilimumab, followed by those who received chemotherapy (median OS: 609 days, 503 days, 392 days, *p* = 0.040, Figure 1A). However, we did not observe a significant difference in PFS between patients with first-line nivolumab and PF, nivolumab and ipilimumab, and chemotherapy (*p* = 0.143, Figure 1B). The longest PFS was observed in patients receiving nivolumab and chemotherapy.

To determine if there was any difference in response to nivolumab administration, we divided the patients receiving nivolumab and chemotherapy based on different lines. We found that there were no differences in the OS and PFS regardless of whether we prescribed nivolumab-containing regimens as first-line, second-line, or after treatment (Figure 2A,B). There is no significant difference of response rate according to different lines of using nivolumab combined with chemotherapy (*p* = 0.575, Table 2). However, a trend of better response was observed whenever immunotherapy is combined with chemotherapy as first-line treatment (Figure 2C).

### 2.3. Liver Metastasis Influenced Dual Immune Checkpoint Inhibitors Treatment Response

In our cohort, we found that patients with liver metastasis had a trend of poor response rates in the entire population (ORR in patients without liver metastasis vs. ones with liver metastasis = 65.45% vs. 25%, *p* = 0.094, Table 3) and showed poor response in dual immune checkpoint blockade treatment (*p* = 0.030, Table 3). We found that patients with initial metastatic disease had better response rates compared to patients with recurrent status at enroll analysis in our entire population (ORR initial metastasis vs. recurrent = 67.34 % vs. 40%, *p* = 0.029, Table 3) but higher CR rates in patients with recurrent status. However, we do not see the impact on response rates in patients who received the I/O-containing regimen.

We did not observe a difference in response rates between the different TMN stages, the locations of esophageal cancer, other metastatic sites, including lung and bone. Patients with poor performance status may not be able to tolerate chemotherapy and related poor outcomes. However, we do not see ECOG impact on treatment response either in all populations or immunotherapy-containing regimens (Table 2 and Table 3).

### 2.4. Program Death Ligand 1 (PD-L1) Expression and Treatment Response

In CheckMate-648 clinical trial, esophageal SCC patients with tumor cell (TC) PD-L1 stain ≥1% had longer progression survival. Because of financial issues, we only picked up six patients who received nivolumab-containing regimens to arrange PD-L1 288-8 pharmDx assay (Agilent/Dako, Glostrup, Denmark). Two of them received first-line nivolumab and PF. Three of them received second-line nivolumab and PF and the rest of them received nivolumab and ipilimumab. One patient who received first-line nivolumab and PF showed TC PD-L1 ≥ 10% and < 50% and reached complete remission (CR) with very long progression-free survival (more than 1400 days). The other patient who received first-line nivolumab and PF showed TC PD-L1 < 1% and reached partial response (PR) with a PFS of 721 days. One patient out of three received second-line nivolumab and PF had TC PD-L1 < 1% and also reached CR with a PFS of 490 days and did not lose response until the last day of follow-up. The other two patients who received second-line treatment received PR. One of them showed TC PD-L1 ≥ 50% and had a PFS of 145 days, but long overall survival after three lines of salvage immunechemotherapy followed by nivolumab maintenance therapy. Another one who showed TC ≥ 5% and <10% had a PFS of 153 days. The only one patient who received nivolumab and ipilimumab showed TC PD-L1 < 1% and did not have a response from immunotherapy. We do not see the impact of PD-L1 expression on response rate (*p* = 0.587, data not showen). The patient who showed PD-L1 TC ≥ 10% had a trend of longer immunotherapy-related PFS (Appendix A).

### 2.5. Adverse Effects between Patients Received Nivolumab Combine with Chemotherapy and Standard Treatment

The most common adverse effect is gastrointestinal (GI) symptoms, including nausea and constipation (Table 4). Patients who received chemotherapy showed a higher incidence of nausea (Nivolumab + C/T vs. Nivo + Ipi vs. PF = 56% vs. 57.14% vs. 85.19%, *p* = 0.02, Table 4). Patients who received nivolumab with chemotherapy (majority of PF) decreased diarrhea symptoms compared with chemotherapy alone (*p* = 0.021, Table 4). There were no adverse endocrine effects in the chemotherapy group and nivolumab with chemotherapy. Regarding hematological adverse effects, anemia and thrombocytopenia showed a significant difference in the studied population, with the highest occurrence in the chemotherapy group patients. Most adverse effects were tolerable. Only one patient who received first-line nivolumab with PF had grade 3 mucositis and grade 3 skin rash and discontinued nivolumab after three cycles of treatment but continued use of PF until six cycles of treatment were completed. This patient achieved a complete response and long-term disease-free survival.

In patients who received first-line nivolumab and PF, second-line nivolumab, and more-than-third-line nivolumab and PF, and nivolumab and ipilizumab, the adverse effects are similar, except for thrombocytopenia. We noticed that thrombocytopenia worsened in second-line-or-more nivolumab and PF patients (Table 5). This may be related to the prolonged use of chemotherapy induced bone marrow suppression. On the other hand, we did not find adverse effects of diarrhea in patients received second-line nivolumab and chemotherapy. However, patients with dual immune checkpoint blockade treatment had highest probability of diarrhea (*p* = 0.044, Table 5).

## 3. Discussion

Due to lifestyle changes, the incidence of esophageal SCC (ESCC) has increased in Taiwan. The prognosis of esophageal cancer is poor, with a 5-year survival rate of 19% in the United States, 12.4% in Europe, and 20.9% in China [15,16,17]. The current standard treatment for locally advanced or metastatic esophageal SCC is chemotherapy, including platinum (cisplatin, carboplatin, and oxaliplatin), 5- fluorouracil (5FU), and taxanes (paclitaxel and docetaxel). However, the treatment response remains poor, with a response rate of 35% to 62%, progression survival of 2.5 to 6.2 months, and median OS of 7.6 to 11.1 months [6].

The expression of PD-L1 in ESCC and the surrounding immune cells is less than 50%. However, it is related to a more advanced stage, including tumors, lymph nodes, and distant metastatic status [18]. High expression of PD-L1 was negatively correlated with the survival of ESCC [8]. Immune checkpoint inhibitors, such as anti-PD-1, bind to PD-1 receptors expressed on the surface of infiltrating cytotoxic T-lymphocytes (CD8+), B-cells, and natural killer (NK) cells in the tumor microenvironment, which enhances the cytotoxic effects of CD8+ T cells [19,20,21,22]. Anti-PD-1 therapies have shown antitumor activity in patients with metastatic esophageal cancer [11,23,24,25]. One study investigating the first-line setting of anti-PD-1 therapy combined with chemotherapy (KEYNOTE 590) enrolled 749 patients, including 73% of patients with ESCC. Pembro + chemo vs. chemo was superior for OS in patients with ESCC CPS ≥ 10 (median 13.9 vs. 8.8 months; HR 0.57; 95% CI, 0.43–0.75; *p* < 0.0001), ESCC (median 12.6 vs. 9.8 months; HR 0.72; 95% CI, 0.60–0.88; *p* = 0.0006) [26]. Checkmate 648 also provided an optimistic data. The study randomly assigned 970 patients to one of three treatment arms, irrespective of PD-L1 expression: (1) nivolumab at 240 mg every 2 weeks plus chemotherapy with fluorouracil and cisplatin every 4 weeks; (2) nivolumab at 3 mg/kg every 2 weeks plus ipilimumab at 1 mg/kg every 6 weeks; or (3) chemotherapy alone. The OS was significantly better among all randomized patients with both nivolumab plus chemotherapy and nivolumab plus ipilimumab compared with chemotherapy alone (13.2, 12.8, and 10.7 months, respectively). Additionally, PFS with nivolumab plus chemotherapy was significantly better than that with chemotherapy alone in patients with PD-L1 ≥ 1% [27]. A meta-analysis that enrolled 4752 advanced esophageal SCC patients from several randomized clinical trials reports that no significant benefit in OS was observed with immunochemotherapy compared with chemotherapy in the subgroup of patients who had a tumor proportion score lower than 1% compared with chemotherapy [28]. Because the PD-L1 detection is not reimbursement by insurance, we only chose six patients who received either first-line immunechemotherapy, second-line immunechemotherapy, or dual immune checkpoint inhibitor treatment to arrange PD-L1 immunostaining. In our limited PD-L1 stain data, we do not see the impact of TC PD-L1 expression on treatment response. However, TC PD-L1 ≥10% seems contribute to the impact on longer immunotherapy-related PFS. One of them who received first-line immunechemotherapy with PD-L1 ≥ 10% reached CR and very long PFS and OS.

In our cohort, patients who received nivolumab with chemotherapy and dual immunotherapy (nivolumab and ipilimumab) had more advanced diseases with more distant metastasis. In our real-world patients, we found a better overall response rate and longer OS in patients receiving nivolumab and chemotherapy. Most of the patients in the chemotherapy group had the chance to receive only one line therapy. The patients in the immunechemotherapy group could receive more lines treatment to prolong treatment response and lifespan. We also noticed that there was no difference of survival between first-line, second-line and third-line nivolumab with chemotherapy treatment. However, patients with dual immunotherapy treatment showed the worst PFS in the immune checkpoint-containing regimen group. We do see that the patients with first-line immunechemotherapy treatment had a higher chance to reach complete remission.

To determine which clinical markers could predict treatment responses, we selected esophageal cancer locations, TMN stages, distant metastatic sites, and recurrent status as parameters to evaluate their impact on treatment response. Only liver and distant lymph node metastasis showed a trend of negative and positive impacts, respectively, on both the whole treatment population and immunotherapy-containing regimen population. However, the sample size is too small to draw a conclusion about the impact of liver metastasis on treatment response. As we know, patients with poor performance status would be related to poor outcome and less tolerable to treatment. In our cohort, we do not see this picture. There was no impact of ECOG on treatment response, no matter standard chemotherapy or immune checkpoint-containing regimen.

Gastrointestinal(GI) adverse effects are the most common ones, followed by mucositis and hematological side effects. Interestingly, patients who received nivolumab combined with chemotherapy showed less GI toxicity, including nausea, diarrhea, and less hematological toxicity, including leucopenia, anemia, and thrombocytopenia, compared with patients who received chemotherapy alone. The majority of patient received standard chemotherapy with cisplatin/carboplatin and 5FU in both groups. Adding nivolumab to chemotherapy seems to reduce GI and hematological toxicity. On the other hand, the adverse effects were similar in patients who received immunochemotherapy as first-line, second-line and third-line treatment. Only diarrhea was more prevalent in patients who received the dual immune checkpoint inhibitor treatment. In our patients, they were tolerable to the immunotherapy-containing regimen treatment.

One limitation of this study is the small sample size, especially in patients who received dual immune checkpoint inhibitor regimens. The choice of treatments other than observation treatment varied in second-line, or more-than-third-line in the chemotherapy group. Because patients need to pay for PD-L1 detection fees and not many patients can afford to do so, we only picked up six patients who received the immunotherapy-containing regimen to check PD-L1 status. The data are too small to analyze.

In our cohort observations, nivolumab combined with standard chemotherapy, platinum, and 5-FU, had higher ORR, longer OS, and a longer PFS, no matter first-line or later on using nivolumab combined with chemotherapy. The PFS and OS were similar regardless of whether the first-line or later-line therapy was used in combination with nivolumab. However, first-line nivolumab combined with PF showed a trend of better treatment response. On the other hand, liver and distant lymph node metastasis showed negative and positive impacts, respectively, on treatment response, including chemotherapy and immunotherapy-containing regimens, but no statistics were significant. TC PD-L1 showed less of an impact on the treatment response of the nivolumab-containing regimen and survival benefit. Patients who received nivolumab and chemotherapy had less GI and hematological adverse effects, compared with those who received chemotherapy.

## 4. Materials and Methods

### 4.1. Patients

We retrospectively analyzed metastatic ESCC patients who received standard systemic chemotherapy (platinum and 5-fluruoracil) and patients who received nivolumab combined with systemic chemotherapy (platinum and 5-fluruoracil, or taxane) as first-line, second-line, third-line, or more-than-third-line treatment (Appendix B), and patients who received double immune checkpoint inhibitors, from 1 October 2017 to 30 June 2021, in EDA hospital. All patients received standard concurrent chemoradiotherapy with platinum and 5-Fluruoracil (PF) during their treatment courses. Radiotherapy was added for recurrent ESCC patients if they could tolerate second radiotherapy according to previous doses and treatment-free periods. We continued follow-up those patients until 30 September 2022.

Based on different treatment choices, we separated patients into three groups: patients who received immunotherapy combined with chemotherapy (group 1), patients who received immunotherapy (nivolumab + ipilimumab) (group 2), and patients who received standard concurrent chemoradiotherapy (group 3). For group 1 patients, we analyzed the outcome for patients who received first-, second-, third-, and more-than-3rd- line therapy. The patient characteristics are shown in Table 1. This study was approved by the Institutional Review Board (or Ethics Committee) of EDA hospital (protocol code EMRP-110-049, EMRP-111-026, and date of approval: 6 July 2021 and 12 October 2022).

### 4.2. Statistics

We used ANOVA to analyze differences in sex, TNM stage, location, metastatic sites, recurrent or primary metastatic diseases, comorbidities, best response rates, and adverse effects between groups 1, 2, and 3, as well as to analyze the response rate differences in different locations of esophageal cancer, TMN stage, and distant metastatic sites. We used the t-test to compare age between groups. Kaplan–Meier survival curve analysis was used to analyze OS and progression-free survival (PFS). SPSS 18.0 (SPSS, Inc.; Chicago, IL, USA) was used to perform the statistical analyses.

### 4.3. Immunohistochemistry Stain of PD-L1

We randomly chose six patients who received nivolumab-containing regimen to arrange PD-L1 288-8 pharmDx assay (Agilent/Dako). Two of them received first-line nivolumab and PF. Three of them received second-line nivolumab and PF and the rest received nivolumab and ipilimumab.

## 5. Conclusions

Unresectable advanced and metastatic esophageal squamous cell carcinoma patients showed better overall survival and progression-free survival when they received nivolumab combined with chemotherapy, compared with standard chemotherapy with/without radiotherapy or dual immune checkpoint inhibitor treatment. Tumor cells PD-L1 expression has no impact on treatment response and survival. The drug’s adverse effects, including GI toxicity and hematological side effects, were decreased during adding nivolumab in chemotherapy.

## Figures and Tables

**Figure 1 ijms-24-07312-f001:**
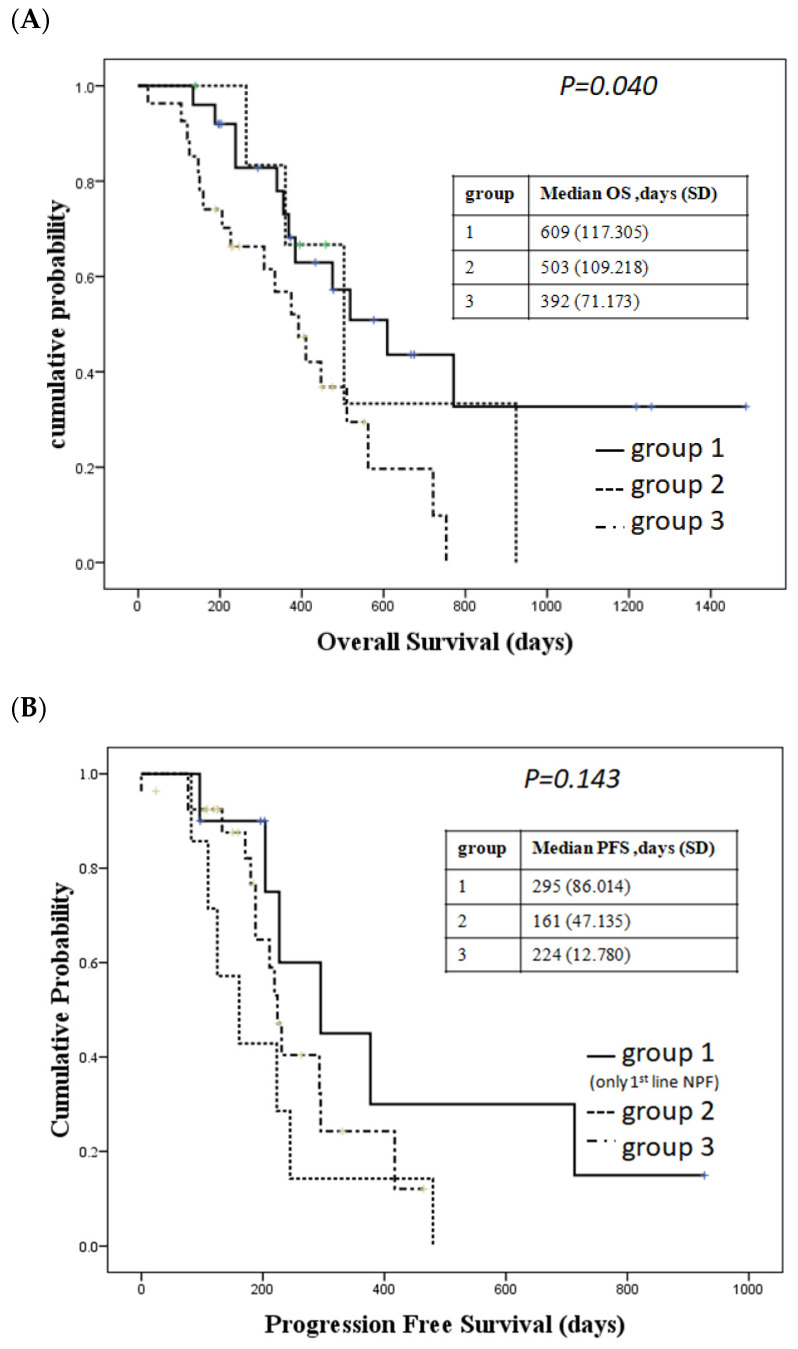
Survival in different treatment groups. (**A**) Overall survival (OS) in different groups. Group 1: patients received nivolumab + chemotherapy; Group 2: patients received nivolumab + ipilimumab; Group 3: patients received chemotherapy with PF as first-line therapy. (**B**) Progression-free survival (PFS). Group 1: patients received nivolumab + chemotherapy as first-line therapy; Group 2: patients received nivolumab + ipilimumab; Group 3: patients received chemotherapy with PF as first-line therapy.

**Figure 2 ijms-24-07312-f002:**
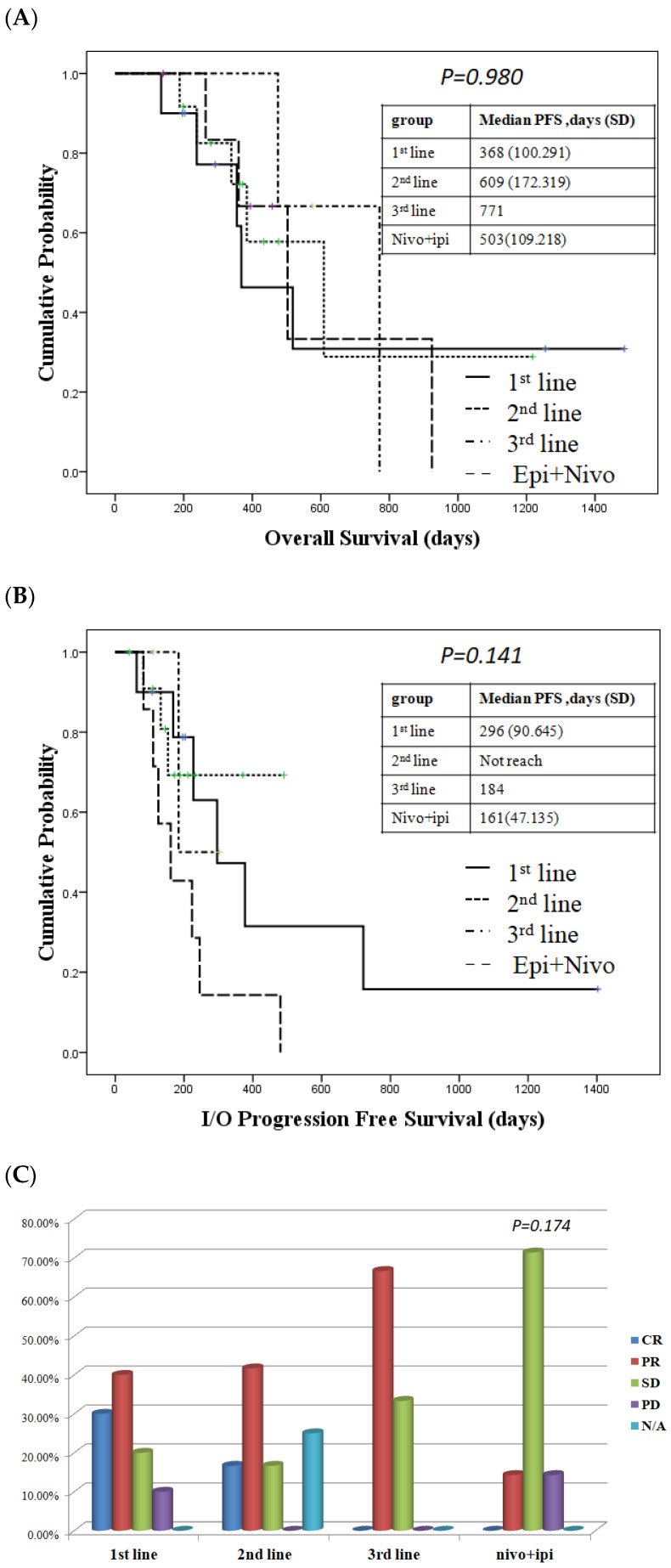
Survival and treatment response in patient who received the immunotherapy-containing regimen. (**A**) Overall survival (OS) in patient who received immunotherapy-containing regimen. (**B**) Progression-free survival (PFS) in patients who received an immunotherapy regimen. The duration of the treatment response was the same regardless of whether patients received first-line, second-line or third-line nivolumab + chemotherapy. (**C**) Response rates in patients who received an immunotherapy regimen.

**Table 1 ijms-24-07312-t001:** Patients’ characteristics.

	Nivolumab + C/T (n = 25)	Nivolumab + ipi (n = 7)	Chemotherapy (n = 27)	*p*-Value
Sex (M/F)	25 (100%)/0	7 (100%)/0	26 (96.30%)/1 (3.70%)	0.547
Age (median ± SD)	53 ± 7.544	56 ± 5.090	59 ± 10.886	0.216
Recurrent	7 (28%)	2 (28.57%)	1 (3.70%)	0.045 *
initial metastasis	18 (72%)	5 (71.42%)	26 (96.30%)	
Stage (AJCC8)				0.020 *
III	2 (8%)	0	2 (7.40%)	
IVA	6 (24%)	0	15 (55.56%)	
IVB	17 (68%)	7 (100%)	10 (37.03%)	
Stage-T 0	5 (20%)	3 (42.85%)	0	0.075
1	0	0	1 (3.70%)	
2	1 (4%)	0	0	
3	9 (36%)	3 (42.85%)	8 (29.63%)	
4	10 (40%)	1 (14.28%)	18 (66.67%)	
Stage-N 0	6 (24%)	2 (28.57%)	3 (11.11%)	0.272
1	9 (36%)	2 (28.57%)	8 (29.63%)	
2	5 (20%)	0	11 (40.74%)	
3	5 (20%)	3 (42.85%)	5 (18.52%)	
Stage-M 0	8 (32%)	0	17 (62.96%)	0.004 *
1	17 (68%)	7 (100%)	10 (37.04%)	
ECOG	22 (88%)	7 (100%)	16 (59.26%)	0.048 *
0	3 (12%)	0	7 (25.93%)	
1	0	0	4 (14.81%)	
2				
Location				
upper	6 (24%)	4 (57.14%)	11 (40.74%)	0.788
middle	7 (28%)	1 (14.28%)	5 (18.52%)	
upper/middle	1 (4%)	0	2 (7.41%)	
Middle/lower	1 (4%)	0	0	
lower	9 (36%)	2 (28.57%)	9 (33.33%)	
panesophagus	1 (4%)	0	0	
Metastatic site (%)				
lung	5 (20%)	3 (42.85%)	3 (11.11%)	0.154
liver	2 (8%)	1 (14.28%)	1 (3.70%)	0.581
bone	5 (20%)	0	2 (7.41%)	0.219
distant LN	5 (20%)	4 (57.14%)	5 (18.52%)	0.092
DM (%)	3 (12%)	2 (28.57%)	4 (14.81%)	0.557
Hypertension (%)	7 (28%)	2 (28.57%)	4 (14.81%)	0.470
Other cancer # (%)	4 (16%)	1 (14.28%)	7 (14.81%)	0.811
Hepatitis B (%)	0	0	1 (3.70%)	0.636
Hepatitis C (%)	0	0	0	
Liver cirrhosis (%)	0	1 (14.28%)	2 (7.41%)	0.355
Best Response (%)				0.038 *
CR	5 (20%)	0	3 (11.11%)	
PR	13 (52%)	1 (14.28%)	15 (55.56%)	
SD	6 (24%)	5 (71.42%)	3 (11.11%)	
PD	1 (4%)	1 (14.28%)	4 (14.81%)	
not measure			2 (7.41%)	
Treatment lines (%)				0.008 *
1	4 (16%)	2 (28.57%)	18 (66.67%)	
2	14 (56%)	3 (42.86%)	8 (29.63%)	
3	5 (20%)	1 (14.28%)	1 (3.70%)	
4	1 (4%)	0	0	
5	1 (4%)	0	0	
6	0	1 (14.28%)	0	
Receive curative esophagectomy	0	0	2 (7.41%)	

# including head and neck cancers, lung cancer, colorectal cancers and biliary tract cancers. * *p* < 0.05.

**Table 2 ijms-24-07312-t002:** Factors related to immune checkpoint inhibitor-containing regimen. I/O response related factors.

	Nivo + Chemotherapy						*p*-Value	Nivo + Epi				*p*-Value
Location (n)	Upper (6)	Upper/Middle (1)	Middle (7)	Middle/Lower (1)	Lower (9)	PANESOPHAGUS (1)		Upper (4)	Middle (1)		Lower (2)	
CR (%)	2 (33)	0	0	0	3 (33.3)	0	0.350	0	0		0	0.478
PR (%)	2 (33)	1 (100)	4 (57)	0	4 (44.4)	0		1 (25)	0		0	
SD (%)	2 (33)	0	1 (14)	0	1 (11.1)	1 (100)		3 (75)	1 (100)		1 (50)	
PD (%)	0	0	0	0	1 (11.1)	0		0	0		1 (50)	
NA (%)	0	0	2 (29)	1 (100)	0	0						
T stage (n)	0 (5)	1 (0)	2 (1)	3 (9)	4a (4)	4b (6)		0 (3)		3 (3)	4a (1)	
CR (%)	1 (20)	0	0	1 (11.1)	1 (25)	2 (33.3)	0.436	0		0	0	0.443
PR (%)	2 (40)	0	1 (100)	5 (55.5)	1 (25)	3 (50)		0		1 (33.3)	0	
SD (%)	1 (20)	0	0	2 (22.2)	1 (25)	1 (16.7)		3 (100)		1 (33.3)	1 (100)	
PD (%)	0	0	0	0	1 (25)	0		0		1 (33.3)	0	
NA (%)	1 (20)			1 (11.1)								
N stage (n)	0 (6)		1 (9)	2 (5)		3 (5)		0 (n = 2)		1 (n = 2)	3 (n = 3)	
CR (%)	2 (33.33)		2 (22.22)	0		1 (20)	0.854	0		0	0	0.380
PR (%)	2 (33.33)		4 (44.44)	3 (60)		2 (40)		0		1 (50)	0	
SD (%)	1 (16.67)		2 (22.22)	1 (20)		1 (20)		2 (100)		1 (50)	2 (66.67)	
PD (%)	0		0	1 (20)		0		0		0	1 (33.33)	
NA (%)	1 (16.67)		1 (11.11)			1 (20)						
M stage	0 (8)			1 (17)				0 (n = 0)			1 (n = 7)	
CR (%)	2 (25)			3 (17.65)			0.606	0			0	
PR (%)	3 (37.5)			8 (47.06)							1 (14.29)	
SD (%)	1 (12.5)			4 (23.53)							5 (71.43)	
PD (%)	1 (12.5)			0							1 (14.29)	
NA (%)	1 (12.5)			2 (11.76)								
Metastatic site (lung) (n)	N (20)			Y (5)				N (n = 4)			Y (n = 3)	
CR (%)	5 (25)			0			0.425	0			0	0.350
PR (%)	8 (40)			3 (60)				0			1 (33.33)	
SD (%)	3 (15)			2 (40)				3 (75)			2 (66.67)	
PD (%)	1 (5)			0				1 (25)			0	
NA (%)	3 (15)											
Metastatic site (liver) (n)	N (23)			Y (2)				N (n = 6)			Y (n = 1)	
CR (%)	5 (21.74)			0			0.464	0			0	0.030*
PR (%)	10 (43.48)			1 (50)				1 (16.67)			0	
SD (%)	5 (21.74)			0				5 (83.33)			0	
PD (%)	1 (4.35)			0				0			1 (100)	
NA (%)	2 (8.70)			1 (50)								
Metastatic site (bone) (n)	N (20)			Y (5)				N (n = 7)			Y (n = 0)	
CR (%)	5 (20)			0			0.425	0			0	
PR (%)	8 (40)			3 (60)				1 (14.29)				
SD (%)	3 (15)			2 (40)				5 (71.43)				
PD (%)	1 (5)			0				1 (14.29)				
NA (%)	3 (15)											
Metastatic site (LN) # (n)	N (20)			Y (5)				N (n = 3)			Y (n = 4)	
CR (%)	2 (10)			3 (60)			0.080	0			0	0.350
PR (%)	11 (55)			0				0			1 (25)	
SD (%)	4 (20)			1 (20)				2 (66.67)			3 (75)	
PD (%)	1 (5)			0				1 (33.33)			0	
NA (%)	2 (10)			1 (20)								
Recurrent $ (n)	N (18)			Y (7)				0 (n = 5)			1 (n = 2)	
CR (%)	3 (16.67)			2 (28.57)			0.796	0			0	0.571
PR (%)	9 (50)			2 (28.57)				1 (20)			0	
SD (%)	3 (16.67)			2 (28.57)				3 (60)			2 (100)	
PD (%)	1 (5.56)			0				1 (20)			0	
NA (%)	2 (11.11)			1 (14.29)								
ECOG	0 (22)			1 (3)				0 (7)			1 (0)	
CR (%)	5 (22.75)			0			0.526	0				
PR (%)	9 (40.91)			2 (66.67)				1 (14.29)				
SD (%)	5 (22.73)			0				5 (71.43)				
PD (%)	1 (4.55)			0				1 (14.29)				
NA (%)	2 (9.10)			1 (33.33)								
Treat line	1st (10)		2nd (12)	3rd (3)								
CR (%)	3 (30)		2 (16.67)	0			0.575					
PR (%)	4 (40)		5 (41.67)	2 (66.67)								
SD (%)	2 (20)		2 (16.67)	1 (33.33)								
PD (%)	1 (10)		0	0								
NA (%)			3 (25)									

# distant lymph node metastasis; $ recurrent.

**Table 3 ijms-24-07312-t003:** Factors related to treatment response.

	All Population ^@^						*p*-Value	I/O-containing Regimen ^&^					*p*-Value
Location (n)	Upper (21)	Upper/Middle (3)	Middle (13)	Middle/Lower (1)	Lower (20)	Panesophagus(1)		Upper(10)	Upper/Middle (1)	Middle (8)	Middle/Lower (1)	Lower (11)	
CR (%)	3 (14.3)	0	0	0	5 (25)	0	0.339	2 (20)	0	0	0	3 (27.3)	0.171
PR (%)	8 (38.1)	2 (66.7)	9 (69.2)	0	10 (50)	0		3 (30)	1 (100)	4 (50)	0	4 (36.4)	
SD (%)	6 (28.6)	0	4 (30.8)	1	2 (10)	1		5 (50)	0	2 (25)	1	2(18.2)	
PD (%)	2 (9.5)	1 (33.3)	0	0	3 (15)	0		0	0	0	0	2 (18.2)	
NA (%)	2 (9.5)	0	0	0	0	0				2 (25)			
T stage (n)	0 (8)	1 (1)	2 (1)	3 (20)	4a (8)	4b (21)		0 (8)	2 (1)	3 (12)	4a (5)	4b (6)	
CR (%)	1 (12.5)	0	0	3 (15)	0	4 (19)	0.438	1 (12.5)	0	1 (8.3)	1 (20)	2 (33.3)	0.343
PR (%)	2 (25)	0	1 (100)	11 (55)	6 (75)	9(42.9)		2 (25)	0	6 (50)	1 (20)	3 (50)	
SD (%)	5(62.5)	0		4 (20)	1 (12.5)	4 (19)		4 (50)	0	3 (25)	2 (40)	1 (16.7)	
PD (%)	0	1 (100)		2 (10)	1 (12.5)	2 (9.5)		0	0	1 (8.3)	1 (20)	0	
NA (%)	0			0	0	2 (9.5)		1 (12.5)	1 (100)	1 (8.3)			
N stage (n)	0 (11)		1 (19)	2 (16)	3 (13)			0 (n = 8)	1(n = 11)	2(n = 5)		3 (n = 8)	
CR (%)	2 (18.2)		2 (10.5)	3 (18.75)	1 (7.69)		0.373	2 (25)	2 (18.18)	0		1 (12.5)	0.848
PR (%)	3 (27.3)		11 (57.9)	8 (50)	7 (53.85)			2 (25)	5 (45.45)	3 (60)		2 (25)	
SD (%)	5 (45.5)		5 (26.3)	1 (6.25)	3 (23.08)			3 (37.5)	3 (27.27)	1 (20)		3 (37.5)	
PD (%)	1 (9.1)		1 (5.3)	2 (12.5)	2(15.38)			0	0	1 (20)		1 (12.5)	
NA (%)	0		0	2 (12.5)	0			1 (12.5)	1 (9.09)			1 (12.5)	
M stage	0 (25)			1 (34)				0 (n = 8)		1 (n = 24)			
CR (%)	3 (12)			5 (14.71)			0.342	2 (25)		3 (12.5)			0.631
PR (%)	14 (56)			15 (44.12)				3 (37.5)		9 (37.5)			
SD (%)	3 (12)			11 (32.35)				1 (12.5)		9 (37.5)			
PD (%)	4 (16)			2 (5.88)				1 (12.5)		1 (4.17)			
NA(%)	1 (4)			1 (2.94)				1 (12.5)		2 (8.33)			
Metastatic site (lung) (n)	N (48)			Y (11)				N (n = 24)		Y (n = 8)			
CR (%)	7 (14.58)			1 (9.09)			0.554	5 (20.83)		0			0.290
PR (%)	23 (47.92)			6 (54.55)				8 (33.33)		4 (50)			
SD (%)	10 (20.83)			4 (36.36)				6 (25)		4 (50)			
PD (%)	6 (12.5)			0				2 (8.33)		0			
NA (%)	2 (4.17)			0				3 (12.5)		0			
Metastatic site (liver) (n)	N (55)			Y (4)				N (n = 29)		Y (n = 3)			
CR (%)	8 (14.55)			0			0.094	5 (17.24)		0			0.113
PR (%)	28 (50.91)			1 (25)				11 (37.93)		1 (33.33)			
SD (%)	13 (23.64)			1 (25)				10 (34.48)		0			
PD (%)	4 (7.27)			2 (50)				1 (3.45)		1 (33.33)			
NA (%)	2 (3.64)			0				2 (6.90)		1 (33.33)			
Metastatic site (bone) (n)	N (52)			Y (7)				N (n = 27)		Y (n = 5)			
CR (%)	8 (15.38)			0			0.303	5 (18.52)		0			0.592
PR (%)	25 (48.08)			4 (57.14)				9 (33.33)		3 (60)			
SD (%)	12 (23.08)			2 (28.57)				8 (29.63)		2 (40)			
PD (%)	6 (11.54)			0				2 (7.41)		0			
NA (%)	1 (1.92)			1 (14.29)				3 (11.11)		0			
Metastatic site (LN) ^#^ (n)	N (45)			Y (14)				N (n = 23)		Y (n = 9)			
CR (%)	4 (8.88)			4 (28.57)			0.120	2 (8.70)		3 (33.33)			0.174
PR (%)	24 (53.33)			5 (35.71)				11 (47.83)		1 (11.11)			
SD (%)	9 (20)			5 (35.71)				6 (26.09)		4 (44.44)			
PD (%)	6 (13.33)			0				2 (8.70)		0			
NA (%)	2 (4.44)			0				2 (8.70)		1 (11.11)			
Recurrent ^$^ (n)	N (49)			Y (10)				N (n = 23)		Y (n = 9)			
CR (%)	6 (12.24)			2 (20)			0.029 *	3 (13.04)		2 (22.22)			0.173
PR (%)	27 (55.10))			2 (20)				10 (43.48)		2 (22.22)			
SD (%)	8 (16.33)			6 (60)				6 (26.09)		5 (55.55)			
PD (%)	6 (12.24)			0				2 (8.70)		0			
NA (%)	2 (4.08)			0				2 (8.70)					
ECOG	0 (45)			1 (10)	2 (4)			0 (29)		1 (3)			
CR (%)	7 (15.56)			0	1 (25)		0.660	5 (17.24)		0			0.338
PR (%)	21 (46.67)			6 (60)	2 (50)			10 (34.48)		2 (66.66)			
SD (%)	12 (26.67)			2 (20)	0			10 (34.48)		0			
PD (%)	4 (8.89)			1 (10)	1 (25)			2 (6.70)		0			
NA (%)	1 (2.22)			1 (10)	0			2 (6.70)		1 (33.33)			

^@^ best response during treatment period; ^&^ I/O-containing regimen response; ^#^ distant lymph node metastasis; ^$^ recurrent: Y. Recurrent at diagnosis at treatment intervention: N. Metastasis or unresectable disease without previous treatment during treatment; NA: not assessment; * *p* < 0.05.

**Table 4 ijms-24-07312-t004:** Adverse effects.

	Nivolumab + C/T (n = 25)	Nivolumab + ipi (n = 7)	Chemotherapy (n = 27)	*p*-Value
GI (all/Gr 3,4, %)				
Diarrhea	5 (20%)/0	4 (57.15%)/0	15 (55.56%)/0	0.021 *
Constipation	14 (56%)/0	6 (85.71%)/0	25 (92.59%)/0	0.842
Nausea	14 (56%))/0	4 (57.14%)/0	23 (85.19%)/0	0.020 *
Vomiting	3 (12%)/0	1 (14.28%)/0	7 (25.93%)/0	0.415
Skin rash (all/Gr 3,4, %)	2 (8%)/1 (4%)	3 (42.85%)/1 (14.28%)	3 (11.11%)/0	0.112
Mucositis (all/Gr 3,4, %)	10 (40%)/3 (12%)	3 (42.85%)/0	9 (33.33%)/0	0.463
Pneumonitis (all/Gr 3,4, %)	8 (32%)/0	3 (42.85%)/0	12 (44.44%)/2 (7.41%)	0.628
Hepatitis (all/Gr 3,4, %)	2 (8%)/0	0/0	7 (25.93%)/3 (11.11%)	0.304
AKI (all/Gr 3,4, %)	2 (8%)/0	0/1 (14.28%)	0/0	0.036 *
Endocrine (all/Gr 3,4, %) #	0/0	1 (14.28%)/0	0/0	0.044 *
Leukopenia (all/Gr 3,4, %)	8 (32%)/1 (4%)	2 (28.57%)/0	17 (62.92%)/7 (25.93%)	0.102
Anemia (all/Gr 3,4, %)	10 (40%)/2 (8%)	2 (28.57%)/1 (14.28%)	23 (85.19%)/2 (7.41%)	0.003 *
Thrombocytopenia(all/Gr 3,4, %)	9 (36%)/2 (8%)	0/0	16 (59.26%)/0	0.018 *
Fatigue(all/Gr 3,4, %)	14 (56%)/0	3 (42.85%)/0	13 (48.15%)/0	0.870

# only analysis I/O groups. * *p* < 0.05.

**Table 5 ijms-24-07312-t005:** Adverse effects in different lines of I/O therapy.

	Nivolumab + C/T, First Line (n = 10)	Nivolumab +C/T Second Line (n = 12)	Nivolumab +C/T ≧ Third Line (n = 3)	Nivolumab + ipi (n = 7)	*p*-Value
GI (all/Gr 3,4, %)					
Diarrhea	3 (30%)/0	0/0	1 (33.3%)/0	4 (57.14%)/0	0.044 *
Constipation	3 (30%)/0	4 (33.33%)/0	1 (33.3%)/0	1 (14.28%)/0	0.830
Nausea	7 (70%)/0	6 (50%)/0	1 (33.3%)/0	3 (42.86%)/0	0.466
Vomiting	2 (20%)/0	0/0	1 (33.3%)/0	1 (14.28%)/0	0.329
Skin rash (all/Gr 3,4, %)	1 (10%)/ 1 (10%)	1 (8.33%)/0	0/0	3 (42.85%)/ 1 (14.28%)	0.351
Mucositis (all/Gr 3,4, %)	4 (40%)/ 2 (20%)	5 (41.66%)/ 1 (8.33%)	1 (33.33%)/0	3 (42.85%)/0	0.715
Pneumonia (all/Gr 3,4, %)	3 (30%)/0	4 (33.33%)/0	1 (33.33%)/0	3 (42.85%)/0	0.930
Hepatitis (all/Gr 3,4, %)	0/0	1 (8.33%)/0	1 (33.33%)/0	0/0	0.632
AKI (all/Gr 3,4, %)	2 (20%)/0	0/0	1 (33.33%)/0	0/1 (14.28%)	0.184
Endocrine (all/Gr 3,4, %) #	0/0	0/0	0/0	1 (14.28%)/0	0.552
Leukopenia (all/Gr 3,4, %)	1 (10%)/0	5 (41.66%)/1 (8.33%)	1 (33.33%)/0	2 (28.57%)/0	0.809
Anemia (all/Gr 3,4, %)	5 (50%)/1 (10%)	3 (25%)/0	1 (33.33%)/0	2 (28.57%)/1 (14.28%)	0.139
Thrombocytopenia (all/Gr 3,4, %)	2 (10%)/0	5 (41.66%)/1 (8.33%)	2 (66.66%)/1 (33.33%)	0/0	0.190
Fatigue (all/Gr 3,4, %)	5 (50%)/0	6 (50%)/0	1 (33.33%)/0	3 (42.85%)/0	0.880

# only analysis I/O groups. * *p* < 0.05.

## Data Availability

Data are unavailable due to ethical restrictions and privacy.

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
