# Peer review of "Immune Checkpoint Inhibitor, Nivolumab, Combined with Chemotherapy Improved the Survival of Unresectable Advanced and Metastatic Esophageal Squamous Cell Carcinoma: A Real-World Experience"

_ijms, 2023, doi:10.3390/ijms24087312_

Round 1

Reviewer 1 Report

An original article with an important role in medical practice. Starting from  the hipothesys that the patients with advanced esophageal squamous cell carcinoma (SCC) have a poor prognosis when treated with standard chemotherapy, the article has a new and original aim: to compare  nivolumab with chemotherapy, dual immunotherapy (nivolumab and ipilimumab), and  chemotherapy with or without radiotherapy.There is a consistent  group of  study subjects , with  a research design appropriate, and  methods adequately described. The conclusions were  supported by the results, so the interest to the readers is high . The quality of presentation is exposed into a literary and medical language  very accessible for the readers.

 The study conclusion is an important  one for the medical practice:nivolumab combined with chemotherapy is a better choice for patients with unresectable esophageal-SCC.

I accept in present form and I recomand this article to the medical doctors for its utility in medical practice.

Author Response

Dear reviewer

Thank you very much.

Reviewer 2 Report

The authors retrospectively analyzed the real-world data of treatment response and survival of patients with metastatic unresectable ESCC treated with nivolumab combined with chemotherapy, compare with dual immune checkpoint inhibitors, and standard chemotherapy.

The study covers some issues that have been overlooked in other similar topics. The structure of the manuscript appears adequate and well divided in the sections. Moreover, the study is easy to follow, but some issues should be improved. Some of the comments that would improve the overall quality of the study are:

I-) Authors must pay attention to the technical terms acronyms they used in the text

II-) Conclusion Section: please add it as separate section from discussion, including some "take-home message".

Author Response

Dear reviewer:

Thank you very much. According to your recommendation, I try to answer as below.

I-) Authors must pay attention to the technical terms acronyms they used in the text

Ans: Thank you very much. We have mention the full name at the first appearance of acronyms in our manuscript.

II-) Conclusion Section: please add it as separate section from discussion, including some "take-home message".

Ans: Thank you for your recommendations. I make a small change in discussion part to “In our cohort observations, nivolumab combined with standard chemotherapy, platinum, and 5-FU, had higher ORR, longer OS, and a longer PFS, no matter 1st-line or latter on to use nivolumab combined chemotherapy………” And Section 5. is our conclusion part , followed the structure of IJMS.